# A Novel Aging-Related Prognostic lncRNA Signature Correlated with Immune Cell Infiltration and Response to Immunotherapy in Breast Cancer

**DOI:** 10.3390/molecules28083283

**Published:** 2023-04-07

**Authors:** Zhixin Liu, Chongkang Ren, Jinyi Cai, Baohui Yin, Jingjie Yuan, Rongjuan Ding, Wenzhuo Ming, Yunxiao Sun, Youjie Li

**Affiliations:** 1Department of Biochemistry and Molecular Biology, Binzhou Medical University, Yantai 264003, China; 2Department of Orthopedics, Qilu Hospital of Shandong University, Jinan 250012, China; 3Department of Pediatrics, Yantai Affiliated Hospital of Binzhou Medical University, Yantai 264100, China

**Keywords:** breast cancer, signature, aging, cancer, nomogram, immunotherapy, tumor mutation burden

## Abstract

Breast cancer (BC) is among the most universal malignant tumors in women worldwide. Aging is a complex phenomenon, caused by a variety of factors, that plays a significant role in tumor development. Consequently, it is crucial to screen for prognostic aging-related long non-coding RNAs (lncRNAs) in BC. The BC samples from the breast-invasive carcinoma cohort were downloaded from The Cancer Genome Atlas (TCGA) database. The differential expression of aging-related lncRNAs (DEarlncRNAs) was screened by Pearson correlation analysis. Univariate Cox regression, LASSO–Cox analysis, and multivariate Cox analysis were performed to construct an aging-related lncRNA signature. The signature was validated in the GSE20685 dataset from the Gene Expression Omnibus (GEO) database. Subsequently, a nomogram was constructed to predict survival in BC patients. The accuracy of prediction performance was assessed through the time-dependent receiver operating characteristic (ROC) curves, Kaplan–Meier analysis, principal component analyses, decision curve analysis, calibration curve, and concordance index. Finally, differences in tumor mutational burden, tumor-infiltrating immune cells, and patients’ response to chemotherapy and immunotherapy between the high- and low-risk score groups were explored. Analysis of the TCGA cohort revealed a six aging-related lncRNA signature consisting of MCF2L-AS1, USP30-AS1, OTUD6B-AS1, MAPT-AS1, PRR34-AS1, and DLGAP1-AS1. The time-dependent ROC curve proved the optimal predictability for prognosis in BC patients with areas under curves (AUCs) of 0.753, 0.772, and 0.722 in 1, 3, and 5 years, respectively. Patients in the low-risk group had better overall survival and significantly lower total tumor mutational burden. Meanwhile, the high-risk group had a lower proportion of tumor-killing immune cells. The low-risk group could benefit more from immunotherapy and some chemotherapeutics than the high-risk group. The aging-related lncRNA signature can provide new perspectives and methods for early BC diagnosis and therapeutic targets, especially tumor immunotherapy.

## 1. Introduction

Breast cancer (BC) is among the most universal malignancies in women worldwide, and its incidence is increasing yearly. It has exceeded lung cancer to become the world’s largest cancer since 2020 [1]. BC is a highly heterogeneous disease, thereby indicating vast differences in gene expression profiles, clinicopathological characteristics, and biological behavior [2]. Tumor stage, histological grade, and molecular subtype are often used as prognostic factors for evaluating BC patients in clinical practice. However, the prognostic information of BC patients cannot be accurately predicted through these clinical characteristics, thereby resulting in inaccurate diagnosis of patient prognosis. Some low-risk patients may be treated unnecessarily or excessively, whereas those with high risk of recurrence or metastasis may be undertreated [3]. Accordingly, it is necessary to utilize novel prognostic biomarkers and a corresponding predictive model to risk-stratify the heterogeneous population with BC and to guide individualized treatment.

Aging is a complex phenomenon caused by many factors, and it is an important and inevitable biological process. It manifests as a gradual loss or degradation of functions at all levels of the human organism [4]. The link between aging and cancer has become stronger in recent years [5]. Aging is a comorbidity of BC, strongly indicating that aging-related transcriptomes contribute to BC progression [6]. Aging-related genes (ARGs) play critical roles in the generation and regulation of senescent cells. They also influence tumor cell progression. Alteration of ARGs may provide novel implications for tumor pathogenesis, diagnosis, and therapy [7]. More and more researchers have demonstrated that aging-related prognostic biomarkers can help obtain the potential diagnostic or prognostic value of malignancies, including colorectal, lung, and ovarian cancers [8,9,10].

Long non-coding RNA (lncRNA) are non-coding RNA with a length greater than 200 nucleotides. Studies have shown that lncRNAs play an important role in the dosage compensation effect, epigenetic regulation, cell cycle and cell differentiation regulation, etc. These lncRNAs have crucial effects in all kinds of biological processes, including cell multiplication, differentiation, invasion, apoptosis, and metastasis [11]. Aging-related lncRNAs (ARlncRNAs) participate in the genesis, metastasis, invasion, chemotherapy resistance, and prognosis of BC [12,13,14,15]. These studies focus on a single lncRNA in BC. A comprehensive signature that includes multiple genes is more predictive than a signature that includes only one gene [16]. A novel BC prognosis model based on aging-related lncRNA has not been reported. Therefore, it is crucial to screen for prognostic ARlncRNAs in BC.

We set up a prognostic model ground on six differentially expressed aging-related lncRNAs and validated their prognostic performance in the GSE20685 dataset. Afterward, we developed a novel nomogram that combines the risk scores with clinicopathological information to provide precise prognostic information for BC patients. Finally, we explored the immune landscape of the tumor micro-environment (TME), analyzed tumor mutational burden (TMB), and predicted the sensitivity to drug therapy with different risk scores (Appendix A). Our findings provided new views to accurately predict the prognosis for BC patients and improve treatment programs.

## 2. Results

### 2.1. Differential Expression of Age-Related lncRNA Identification

First, 1135 ARlncRNAs were appraised on account of co-expression analysis of ARGs and lncRNAs expression levels in BC samples (|Pearson R| > 0.4, *p* < 0.001). We screened 287 significant DEarlncRNAs between 1022 BC patients and 112 normal patients for subsequent analysis (|log_2_ FC| > 1 and *p* < 0.05). We incorporated 198 up-regulated genes and 89 down-regulated genes (Figure 1A; Appendix A).

### 2.2. The Construction of Age-Related lncRNA Signature and Verification

First, 64 prognostic lncRNAs related with aging were appraised in the training set on account of the univariate Cox regression analysis. Second, we implemented the LASSO regression to reduce overfitting. The quantity of aging-related lncRNAs was reduced to 20 (tenfold cross validation (10-CV) gained prognostic lncRNAs by minimum lambda values)(Figure 1B,C). Subsequently, a risk signature composed of six lncRNAs was established by the multivariate COX analysis, including MCF2L-AS1, USP30-AS1, OTUD6B-AS1, MAPT-AS1, PRR34-AS1, and DLGAP1-AS1 (Appendix A). In the light of the risk score formula, we sequentially scored the risk of TCGA-BRCA patients. Risk Score = MCF2L-AS1 × (−0.6581) + USP30-AS1 × (−0.9132) + OTUD6B-AS1 × (1.4151) + MAPT-AS1 × (−0.8676) + PRR34-AS1 × (−1.0859) + DLGAP1-AS1 × (−1.5074). Based on the optimal X-tile truncation of risk score, the training set was split into low-risk and high-risk groups. In the training cohort, the optimal cutoff was identified as −6.6. Furthermore, between the two risk subgroups, the heatmap presented the expression pattern of the six lncRNAs related with aging (Figure 1F). The Sankey diagram indicated the correlation among risk types, genes related with aging, and prognostic lncRNAs related with aging (Figure 1D). The co-expression network between these six lncRNAs and genes related with aging in BC was visualized using Cytoscape 3.7.2 software (Figure 1E). Compared with the subgroup with low risk, the subgroup with high risk had observably lower OS, which was indicated by the Kaplan–Meier analysis. In the two different risk groups, BC patients with higher risk score showed lower overall survival (Figure 1H). Furthermore, the first-rank predictability of the prognosis risk score in BC patients with AUCs of 0.753, 0.772, and 0.722 in 1, 3, and 5 years, respectively, was proved by the ROC curve (Figure 1G). To prove the precision of the aging-related lncRNA feature, every patient in the test cohort was given a risk score on the same basis as the training queue formula. The testing cohort patients were then split into two different risk subgroups employing an identical cutoff value (Figure 2A). On the basis of the training cohort, risk scores can distinguish BC patients from two different risk subgroups, and the lower the risk score was, the better prognosis the BC patient had (Figure 2C). The satisfactory predictability of the BC patients’ prognosis with AUCs of 0.662, 0.646, and 0.637 in 1, 3, and 5 years, respectively, was demonstrated by the risk score (Figure 2F).

We downloaded the GSE20685 dataset, but we excluded patients with a lack of clinical information. Finally, we had 327 BRCA patients (Appendix A). The TCGA-BRCA cohort was split into a low-risk subgroup and a high-risk subgroup on the basis of the same cutoff as the TCGA-training (Figure 2B). Compared with the subgroup with low risk, the OS of the subgroup with high risk was remarkably lower, which was indicated by the Kaplan–Meier test (Figure 2D). The AUCs at 1, 3, and 5 years were 0.719, 0.712, and 0.680, respectively (Figure 2G). The expression values of prognostic lncRNAs of BC patients by the external validation cohort was extracted, and the risk score of 327 patients in the GSE20685 was calculated on the basis of the risk score formula. Patients in the GSE2068 cohort were split into two different subgroups in the light of the truncation value of the training queue (Appendix A). We validated the reliability of the aging-related lncRNA signature in predicting prognosis through the ROC curve (Figure 2E,H). The protein levels of these eight aging markers were used to express normal and BC tissues (Figure 3).

### 2.3. Construction of a Nomogram of BC Patients

Then, we assessed whether a signature-based risk score was a separate prognostic factor for BC patients. According to the univariate Cox regression analysis, risk score was a prognostic indicator for BC patients (HR = 1.46, 95% CI = 1.30–1.64, *p* < 0.001) (Figure 4A). The risk score remained a separate prognostic marker for OS (HR = 1.47, 95% CI = 1.31–1.65, *p* < 0.001) in the multivariate Cox regression analysis, which adjusted for other clinical characteristics (Figure 4B). Simultaneously, the ROC curve—which is time-dependent—indicated that, compared with other clinical characteristics, the AUS of the risk score in 1, 3, and 5 years achieved 0.713, 0.707, and 0.673, respectively (Figure 4C–E). Compared with other independent clinical characteristic variables, the risk score was better. To further improve the clinical application of the aging-associated lncRNA signature, we formed a novel nomogram that combined the clinicopathological information with risk score to forecast OS at 1, 3, and 5 years in BC patients (Figure 4F). The ROC curve demonstrated excellent predictability, with AUCs at 1, 3, and 5 years reaching 0.863, 0.805, and 0.771, respectively (Figure 4H). The predictive performance and clinical value of the prognostic chart was evaluated by calibration curve and DCA analyses. The predicted possibility of 1-year, 3-year, and 5-year OS in the nomogram was in accord with the actual observation, which was indicated by the calibration curve (Figure 4G). This nomogram had compelling specificity and sensitivity in predicting clinical outcomes. The DCA curves confirmed our expectations. The nomogram had optimal predictive net benefit compared with traditional clinicopathological features (Figure 4J). Meanwhile, the C-index proved that the prognostic nomogram had better discrimination ability (Figure 4I). These abovementioned outcomes demonstrated that, for predicting the prognosis in BC patients clinically, the nomogram was appropriate.

### 2.4. Correlation of the Risk Score with Clinicopathological Features

We evaluated the correlation between the risk score and clinicopathologic features in BC patients. The low-risk group highly expressed MCF2L-AS1, MAPT-AS1, USP30-AS1, PRR34-AS1, and DLGAP1-AS1, whereas the high-risk group highly expressed OTUD6B-AS1, as indicated in the heat map (Figure 5A). The risk score showed different patterns of distribution among different clinicopathological features of BC patients, involving TNM staging, survival status, age, and clinical stage (Figure 5B,C; Appendix A). Risk scores were higher in BC patients with deeper invasion (stage T3-T4), advanced stage (stage III-IV), or who had died (Figure 5B,C). We also conducted survival analyses of aging features in different clinical feature subgroups. The BC patients had a higher probability of survival across age, M stage, T stage, and clinical stage in the low-risk group. Overall, the aging-associated model was well-correlated with clinical features and was able to predict survival in BC patients who had different clinical features (Figure 5D–J). We explored the independent prognostic value of the six molecular subtypes while considering the molecular heterogeneity of BC. The use of the Kaplan–Meier curve was assessing the connection between the expression of the six lncRNAs in the prognosis of BC patients and our risk model. In BC patients, only the high expression of OTUD6B-AS1 was linked with poor prognosis, whereas in BC patients, the high expression of the other five lncRNAs was linked with a good prognosis (Figure 5K–P). The risk score was inseparable from the clinicopathological characteristics of BC and can be used as an efficient auxiliary tool for forecasting the prognosis of BC patients, which these results demonstrated.

### 2.5. Differential Analysis of Signaling Pathway and Immune Functions in the Groups with High Risk and Low Risk

We utilized GSEA to investigate the discrepancies in the KEGG and GO between the two groups. In the subgroup with high risk, the enriched signaling pathways were linked with tumor initiation and progression, such as PPAR signaling pathways, ECM receptor interactions, and TGF-β signaling pathways, which the KEGG analysis proved (Figure 6A). The high-risk subgroup was enriched with genetically altered cellular components, such as chromosomal region chromosomes and centromeric regions, as shown in the GO analysis (Figure 6C). In addition, the enriched signaling pathways in the group with low risk, such as Graft-versus-host disease, autoimmune thyroid disease, and primary immunodeficiency, were related to immunity according to the KEGG analysis (Figure 6B). The GO analysis indicated enriched biological processes of mRNA splice junction alterations, such as mRNA splice site selection and mRNA five splice site recognition (Figure 6D). Considering the variety of enriched immune-related pathways, we performed a detailed quantification of 16 immune cells and the corresponding 13 immune pathways and functions by the ssGSEA. We further searched the correlation between the different risk score and immune pathways. Five immune functions and six immune cell types were obviously linked with the aging-related risk score (Figure 6E,F). Moreover, the PCA revealed that, contrasted with all genes (Figure 7A) and all lncRNAs (Figure 7B), the BC patients were obviously divided into two different groups by the six lncRNAs in the risk model (Figure 7C).

### 2.6. Analysis of Immune Cell Infiltration

We further explored whether the aging-related lncRNA signature was connected with tumor immunity. We employed the XCELL, QUANTISEQ, EPIC, TIMER, and MCPCOUNTER immune databases to calculate the differences between high- and low-risk subgroups in the proportion of tumor-infiltrating immune cells for the aging signature (Figure 7D). The relative proportions of the 22 tumor-infiltrating immune cells were further analyzed by the CIBERSORT in each sample. Interestingly, in the two different risk groups, the enrichment of the tumor-infiltrating immune cells showed different dimensions (Figure 8B; Appendix A). Subsequently, the connection of the tumor-infiltrating immune cells and risk score was explored through Spearman’s analysis [17]. Figure 8C,D indicates that resting NK cells, M2 macrophages, and M0 macrophages were positively related with risk score. Plasma cells, B memory cells, monocytes, activated NK cells, resting mast cells, regulatory T cells, and CD8+ T cells were all negatively associated with risk score. Meanwhile, eight immune cells were remarkably related with the risk score on the basis of the Pearson correlation analysis (Figure 8E–L). The outcomes showed that the six lncRNAs in our risk model can discriminate different features of immune cell infiltration in BC.

### 2.7. Genetic Alteration in Aging-Related Genes

We examined the prevalence of somatic mutations and CNVs in entire genes of BC and discovered that the most frequent variant class was missense mutations, whereas single-nucleotide polymorphisms (SNPs) were the most common variant types. The highest SNV classification was C > T (Appendix A). Then, the frequency and class of mutations in total genes were investigated in two different risk groups. In Figure 9A,B, genetic mutations were found in 247 of 308 (80.2%) BC samples in the low-risk subgroup and 714 of 714 (100%) BC samples in the high-risk subgroup. The most frequent variant class was missense mutations. Among high-risk subgroups, the mutation rate of PIK3CA was high (33.9%) and was second only to the mutation rate of TP53 (44.4%). The PIK3CA had the most alterations (34%) in the subgroup with low risk (Figure 9A,B). Additionally, when we figured the TMB for every BC patient, we discovered that, in the high-risk group, TMB was remarkably higher (Figure 9C). However, no association was observed between TMB and OS in BC patients (Figure 9D).

### 2.8. Immunotherapy Effect and Drug Sensitivity

Immune checkpoint inhibitors (ICIs) are currently approved for the therapy of PD-L1+ metastatic triple-negative BC [18]. Between high-risk and low-risk groups, discrepancies in the expression of immune molecules related with checkpoint were investigated. In the low-risk group, the expression of all-important immune checkpoint-related molecules was up-regulated according to the result (Figure 8A). This provided a potential immunotherapy target for BC patient whose risk scores were different. Subsequently, the use of IPS was predicting the reaction of different BC patients to ICI. In order to gauge how BC patients in the two separate risk categories responded to anti-PD-1/PD-L1 and anti-CTLA-4 medication, two subtypes of IPS values—IPS-PD-1/PD-L1 pos and IPS-CTLA-4 pos—were utilized [19]. The immunophenoscore (IPS) of the TCGA-BRCA cohort was downloaded from the Cancer Immunome Atlas (TCIA) database. The form downloaded from the official website (https://tcia.at/home, accessed on 13 January 2023) already defines the positive and negative information for each patient. IPS scores in the low-risk group were higher than those in the high-risk group (Figure 9E). The outcomes showed that the immunotherapy effect was better in the low-risk group. Based on the potential role of lncRNAs in modulating drug sensitivity, we assessed the potential of the 6-ARlncRNAs signature as biomarkers for predicting drug response in BC patients (Figure 10). In TCGA-BRCA patients, we deduced the IC50 values for 138 medications. Patients in the high-risk group may react to Nutlin more strongly than Temirolimus, 3a, and other drugs. BC patients in the low-risk group may be more susceptible to drugs such as AZD6482 and Thapsigargin.

## 3. Discussion

BC is the primary cause of cancer-related death in women worldwide, and the incidence rate is rising yearly. Despite improvements in diagnosis and therapy, invasive BC’s high death rate remains a global concern [20]. Clinical consequences are highly variable in patients, which is likely due to BC’s heterogeneity [21]. Therefore, it is urgent to search for and evaluate prognostic biomarkers for the early diagnosis of BC. The lncRNAs can promote tumor initiation and progression and can serve as biomarkers to predict the prognosis of cancer patients [22,23]. Aging is an important and unavoidable biological process; it leads to the progressive deterioration of the function of many tissues [24]. Studies have pointed out that ARGs can promote tumor initiation, progression, and metastasis. Cancer may be inhibited by managing senescence in tumor cells [25,26]. Aging is an independent risk factor for some cancers [24]. In addition, aging-related markers have prognostic potential for predicting cancer [27]. The above facts indicated that we urgently need to identify more aging-related molecular markers in BC patients.

A prognostic model of aging-related lncRNAs including six lncRNA molecules (MCF2L-AS1, USP30-AS1, OTUD6B-AS1, MAPT-AS1, PRR34-AS1 and DLGAP1-AS1) was constructed. Most of the aging-related lncRNAs included in this prognostic signature are closely linked to tumorigenesis, metastasis, and proliferation. MCF2L-AS1 promotes colorectal cancer development through the miR-105-5p/RAB 22A axis and regulates miR-873-5p levels to promote cancer stem-like features of non-small-cell lung carcinoma cells [28,29]. However, whether MCF2L-AS1 plays a role in BC has not been reported. USP30-AS1 was silenced to promote mitochondrial uncoupler-induced mitophagy as a negative regulator of mitochondrial homeostasis. Mitochondrial autophagy leads to decreased mitochondrial function, a hallmark of cancer [30]. For some malignant tumors, USP30-AS1 is a potential prognostic biomarker [31,32]. Multiple tumor types include the lncRNA ovarian tumor domain that contains 6B antisense RNA1 (OTUD6B-AS1). OTUD6B-AS1 has also been discovered to be a new tumor-associated lncRNA [33]. Several reports are consistent with our view that OTUD6B-AS1 can serve as a predictor of BC prognosis [34,35]. MAPT-AS1 can affect BC proliferation, migration, and drug sensitivity [36,37]. PRR34-AS1 was up-regulated in hepatocellular carcinoma and pediatric medulloblastoma [38,39]. However, no reports explored the role of PRR34-AS1 in BC. According to a recent study, DLGAP1-AS1 may activate the Wnt signaling pathway to speed up the proliferation of glioblastoma and hepatocellular cancer [40,41]. However, the relationship between BC and DLGAP1-AS1 has not been pointed out. Using a ROC curve, we assessed the prognostic model composed of these six IncRNAs’ capacity to forecast the prognosis of BC. This model did well in prediction (AUCs of 0.753, 0.772, and 0.722 for 1, 3, and 5 years, respectively). We built a novel nomogram associated with the risk score and clinical characteristics according to the lncRNA signature. The resulting verification demonstrated the compelling sensitivity and specificity of the nomogram in predicting clinical results. Correlation between clinical characteristics and the aging signature in BC patients was evaluated. The risk score correlated well with tumor clinical stage (T and M stages) and age, with the low-risk group showing better survival. This aging model indicated better predictive performance for early-stage BC patients. Therefore, this prognostic model enables better risk stratification for early BC patients.

The immune response in tumors can be triggered by aging. The tumor-infiltrating immune cell microenvironment contributes to tumor development [42]. For example, cytotoxic T, NK, and B cells disrupt tumor cells, whereas myeloid-derived suppressor cells (MDSCs), tumor-associated macrophages (TAMs), and regulatory T cells (Tregs) coordinate tumor growth and immune escape [43]. The relationship between aging and the immune cells that infiltrate tumors, however, was rarely covered in studies. The percentage of immune cells that infiltrated the tumor was determined using the CIBERSORT algorithm analysis to see whether an aging-related signature is connected to tumor immunity and immunotherapy. The prognostic model can recognize the different features of tumor-infiltrating immune cells well. Primitive B cells, regulatory T cells, and CD8+ T cells—tumor-killing immune cells—were less prevalent in the high-risk group than in the low-risk group, but M0 and M2 macrophages, which encourage tumor growth and progression, were more prevalent [44]. Five immune functions and six immune cell types were substantially linked with the aging-related risk score, according to the ssGSEA. After that, we looked into variations in immune checkpoint molecule expression between groups at high risk and those at low risk. The outcomes showed that the expression of all important immune checkpoints, including CTLA-4 and PD-1, was up-regulated in the low-risk group, indirectly suggesting that the low-risk group had pre-existing T cell activation. Therefore, BC patients who had a low risk score may benefit more from ICI treatment. Microsatellite instability, TMB, PD-L1 expression, and mismatch repair deficiency are used for patient selection before ICI therapy [45]. In this study, TMB was higher in high-risk BC patients, but no significant association was found between TMB and OS in BC. In addition, the IPS quantitatively predicts patient response to anti-PD-1/PD-L1 and anti-CTLA-4 therapies [46]. The IPS score of the low-risk group was higher than that of high-risk group (*p* < 0.001). The 6-ARlncRNAs signature can be used for patient selection before ICI treatment, and ICI treatment is more appropriate for BC patients with lower aging-related risk scores. To predict possible drug targets, analysis was conducted. The IC50 of Nutlin.3a, Temsirolimus, and others in the high-risk group are higher than those in the low-risk group, i.e., higher sensitivity, whereas the patients who are in the low-risk group are more sensitive to AZD6482, Thapsigargin, and others. Guidance was provided for the selection of therapeutic drugs.

In previous studies, some aging-related molecular markers that can predict invasive BC have been verified [27], but the utilization of aging-related lncRNAs in BRCA has not been presented. We utilized many samples in the TCGA and GEO databases to identify and validate the risk model, which can predict the survival, pathological characteristics, and treatment methods of BC patients well, thereby providing powerful guidance for clinical practice. The data utilized in our research came from several public databases. The 6-ARlncRNAs signature offers a fresh viewpoint on BC diagnosis, prognosis, and treatment, but further clinical trials are required to confirm the therapeutic relevance of these results.

## 4. Materials and Methods

### 4.1. Data and Clinical Information Acquisition and Collation

Transcriptome expression data and clinical data of BC patients were downloaded through the Gene Expression Omnibus (GEO) database (https://www.ncbi.nlm.nih.gov/geo/, accessed on 13 January 2023) (National Center for Biotechnology Information, NCBI) and The Cancer Genome Atlas (TCGA) database (https://portal.gdc.cancer.gov/repository, accessed on 13 January 2023) (National Cancer Institute, NCI; National Human Genome Research Institute, NHGRI). Patients with missing clinical information were removed. We then downloaded the gene transfer format (GTF) file through the Ensembl database (http://asia.ensembl.org, accessed on 13 January 2023) (Wellcome Sanger Institute). Transcriptome sequencing data of mRNAs and lncRNAs were annotated and differentiated [47]. We obtained a total of 279 ARGs through the Human Aging Genome Resource (HAGR, https://genomics.senescence.info/cells/, accessed on 13 January 2023) (senescence.info). The ARGs are shown in Appendix A.

### 4.2. Identification and Differential Analysis of Aging-Related lncRNAs

We sifted the differential expression of aging-related lncRNAs (DEarlncRNAs) by using Pearson correlation analysis [48]. Screening criteria were |Pearson R| > 0.4 and *p* < 0.001. Filtering of DEarlncRNAs was performed through the R package “limma” with filter criteria of |log_2_Fold Change| > 1 and FDR < 0.05 [49].

### 4.3. Construction and Validation of the Aging-Related lncRNA Prognostic Signature

We obtained 1022 BC samples and 112 paired normal samples from TCGA database. These BC patients were randomized 1:1 to the training set (n = 511) or the test set (n = 511). Meanwhile, there was no statistical difference in the clinical characteristics of BC patients between the training subgroup and the test subgroup (*p* > 0.05). The “forestplot,” “survminer,” and “survival” packages in R were employed to execute the univariate and the multivariate Cox regression analysis [50]. First, we screened DEarlncRNAs associated with overall survival (OS) in the TCGA breast cancer (BRCA) cohort by using the univariate Cox proportional hazard regression analysis. The LASSO–Cox regression analysis compressed differentially relevant regulators of prognosis, and we removed redundant genes by using the R package “glmnet” [51]. Afterward, the multivariate Cox analysis and stepwise variables were employed to further screen variables associated with OS in the TCGA-BRCA cohort, and we established a predictive signature [52]. The Akaike information criteria were the basis for the signature building. The TCGA-BRCA patients were scored according to the risk scoring formula. Risk score = coef gene (1) × exprgene (1) + coef gene (2) × exprgene (2) + … + coef gene (n) × exprgene (n) (8). We obtained coefficients of each gene by multivariate Cox regression analysis. On the basis of the best cut-off value confirmed by X-tile software, the TCGA-BRCA cohort was split into high- and low-risk subgroups. Afterward, we applied the “survival” R package and compared the diversity in OS in the two subgroups by the log-rank test. The R “pheatmap” package is used to visualize the allocation of corresponding risk values between the two groups [53]. The “timeROC” package was used to obtain receiver operating characteristic (ROC) curves to determine the accuracy of prediction for 1-, 3-, and 5-year OS of BC. The prediction accuracy was judged by the area under the curve (AUC) value [54].

The GSE20685 dataset was downloaded through the GEO database. A total of 327 patients with BRCA were included. The GSE20685 dataset was employed as an external validation set. We performed batch-to-batch correction of gene expression values between the GSE20685 cohort and the TCGA-BRCA cohort to eliminate batch differences with the ComBat function and the R “sva” package [55]. Afterward, the GSE20685 cohort was also split into the two different risk subgroups on the basis of risk cutoff values in the training cohort. The Kaplan–Meier survival curve was also drawn to assess differences of OS in the two different risk subgroups. The ROC curve assessed the ability to predict prognosis in the GSE20685 cohort. Afterwards, protein expression levels of aging-related markers (including p21, p16, p53, TNF-α, IL-6, FGF, VEGF, and MMP-3) were compared between BC and common samples from the Human Protein Atlas (HPA) database (https://www.proteinatlas.org/, accessed on 13 January 2023) (Knut and Alice Wallenberg Foundation) [56].

### 4.4. Development and Evaluation of a Nomogram in BC Patients

We employed the R package “rms” to build a nomogram by combining an aging-related signature with clinical characteristics to predict survival in BC patients [57]. The total score was summed up by each point that corresponded to each variate in the nomogram scoring system to forecast 1-, 3-, and 5-year OS in patients with BC. Meanwhile, we plotted the calibration curve by using the “bootstrap” package. The decision curve analysis (DCA) was drawn by the R package “rmda” to assess the precision of the nomogram [58]. In addition, the consistency index (C-index) and the ROC curve over time were yielded.

### 4.5. Kyoto Encyclopedia of Genes and Genomes, Gene Ontology, Gene Set Enrichment Analysis and Single-Sample Gene Set Enrichment Analysis

First, we carried out the Kyoto Encyclopedia of Genes and Genomes (KEGG) and the Gene Ontology (GO) functional enrichment analyses based on DEarlncRNAs with the packages “GOplot” and “KEGGplot” in R [59]. Second, employing GSEA-3.0 software (http://software.broadinstitute.org/gsea/index.jsp, accessed on 13 January 2023), we performed the Gene Set Enrichment Analysis (GSEA) to show the potential pathways and mechanisms of two different subgroups in the TCGA-BRCA cohort [60]. The GO and KEGG analyses were conducted and visualized through the packages “enrichplot” and “lusterProfiler” in the aging-related signature [61]. In addition, immune infiltration of 13 immune-related pathways and 16 cell types in two different risk subgroups was evaluated employing single-sample genome enrichment analysis (ssGSEA) using the package “GSVA” [62].

### 4.6. Immune Cell Infiltration Analysis

We assessed the scores of 22 immune cell types in the high- and low-risk subgroups in the TCGA-BRCA cohort by using the CIBERSORT algorithm, and the results were filtered (*p* < 0.05) [63]. Through the R package “pheatmap,” differential tumor-infiltrating immune cells between the two subgroups were mapped. Afterward, the association between the proportion of tumor-infiltrating immune cells and the risk score was compared based on the Spearman rank correlation test in TCGA-BRCA cohort [64]. In addition, discrepancies in the proportion of immune cell infiltration in the groups were analyzed based on QUANTISEQ, XCELL, EPIC, TIMER, and MCPCOUNTER immunization databases. Moreover, the principal component analysis (PCA) was applied for grouping visualization of high-dimensional data, model identification, and effective dimensionality reduction. We used the R package “stats” for analysis, Z-Score on the expression spectrum, and further dimensionality reduction analysis using the PRCOMP function to obtain the dimensionality reduction matrix.

### 4.7. Immunotherapy and Drug Sensitivity Prediction

We evaluated the differential expression of immune-checkpoint-related molecules. Different expressions of immune checkpoints in the prognostic risk model can be applied to provide new treatment ideas and predict the clinical treatment effect of corresponding inhibitors. Afterward, the immunophenoscore (IPS) of the TCGA-BRCA cohort was downloaded from the Cancer Immunome Atlas (TCIA) database (Institute of Bioinformatics). Tumor immunogenicity was positively correlated with the IPS value. IPS predicts patient response to immune checkpoint inhibitor (ICI) therapy [46]. Data of IPS were subsequently extracted for analysis. We downloaded the single nucleotide variation dataset for TCGA-BRCA patients from the TCGA database. According to the six lncRNAs in our risk model, we used the R package “maftools” to explore copy number variations (CNVs), somatic mutation, and TMB in the two different risk subgroups [65]. We calculated the TMB of each BC patient to compare differences in the risk subgroups (mutations per million bases). To assess drug susceptibility, we downloaded an anticancer drug dataset from the Genomics of Cancer Drug Sensitivity (GDSC) database (Wellcome Sanger Institute, Genome Research Limited). We deduced the half maximal inhibitory concentration (IC50) values for 138 drugs using the pRRophetic algorithm [66]. Finally, the Wilcoxon test of these drugs was conducted to predict the level of response to the drug in patients at different risk levels.

### 4.8. Statistical Analysis

R is a language for statistical analysis and drawing, and it is an excellent tool for statistical calculation and statistical drawing. Its full name is The R Programming Language. We used R (R Programming Language version 4.1.2) with associated R package for all statistical analyses and plots (Institute for Statistics and Mathematics of WU). For comparison between the two groups, if the normal distribution is met, a *t*-test is used. If it is a non-normal distribution, the nonparametric test is used. The Pearson correlation analysis was employed to assess correlation. The log-rank test, Cox regression, and Kaplan–Meier curves were applied to evaluate prognostic value. The Wilcoxon rank sum test was applied to evaluate the relationship between immune checkpoints, TMB, immune cell infiltration, and IC50 values of chemotherapeutic drugs between these two groups. All statistical analyses were two-sided, and *p* < 0.05 was considered statistically significant if not specified.

## 5. Conclusions

We created a brand-new lncRNA predictive signature for aging. The training set was used to build the signature. It was well-validated on the validation set and has the potential to be a predictive biomarker for patients with invasive BC. The 6-ARlncRNAs signature can offer fresh perspectives and approaches for early diagnosis and therapeutic target identification of BC, including tumor immunotherapy. This research provided a new reference for further research on aging, tumor immunity, and chemotherapy drug selection.

## Figures and Tables

**Figure 1 molecules-28-03283-f001:**
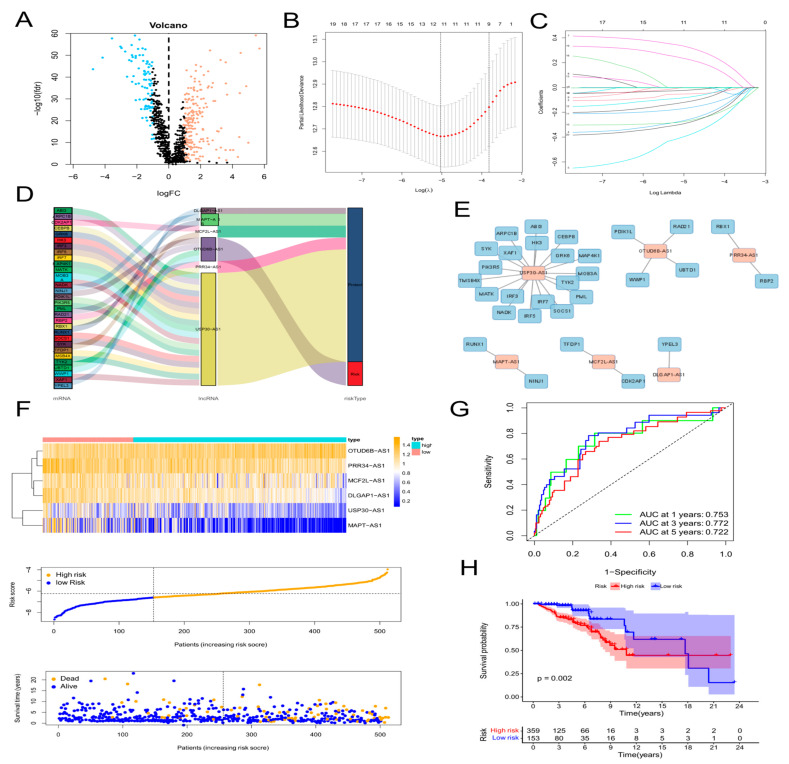
Construction of an aging-related lncRNA prognostic signature. (**A**) Volcano plot of differentially expressed lncRNAs in tumor samples and normal samples using the Pearson correlation analysis. Red dots represent up-regulated genes and blue dots represent down-regulated genes. (**B**,**C**) LASSO analysis to screen 20 prognostic aging-related lncRNAs by tenfold cross validation. (**D**) Sankey diagram of the relationship of 6 aging-related lncRNAs with related mRNAs and risk types. (**E**) Construction and visualization of co-expression network of aging-related lncRNAs and mRNAs using Cytoscape. Red rectangles represent prognostic lncRNAs, and blue rectangles represent aging-related mRNAs. (**F**) Expression patterns of 6 aging-related genes, relationship between risk scores and survival times, and distribution of risk scores in high- and low-risk-score groups. (**G**) Time-dependent receiver operating characteristic (ROC) curves and area under the curve (AUC) analyses in the TCGA training cohort. (**H**) Kaplan–Meier curves of overall survival of BC patients in high-risk and low-risk score groups by the log-rank test.

**Figure 2 molecules-28-03283-f002:**
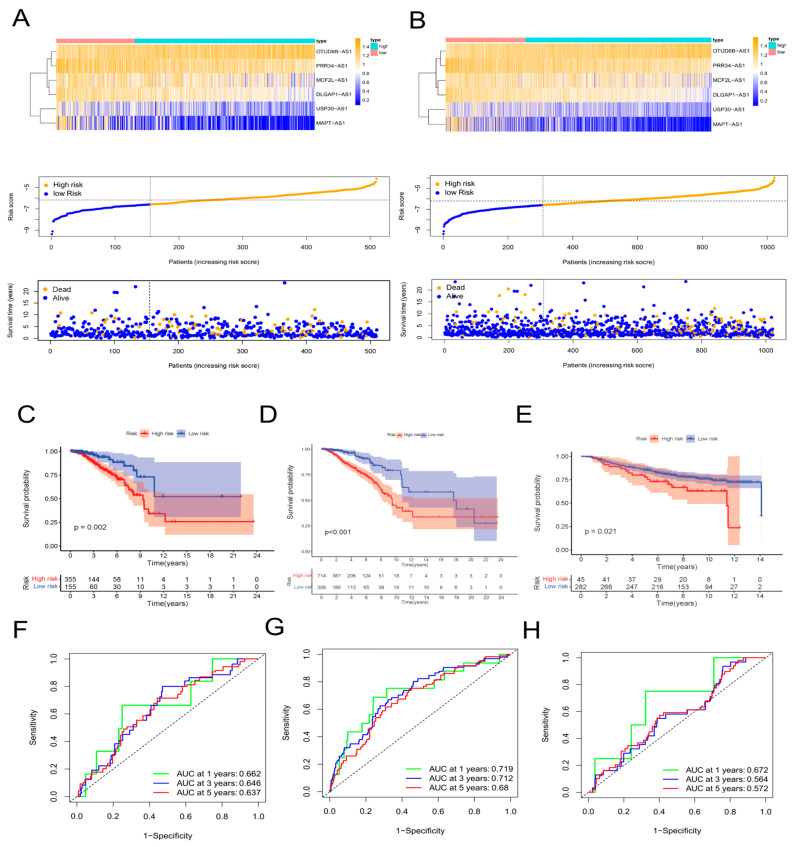
Validation of the aging−related lncRNA prognostic signature. Expression patterns of 6 aging−related genes, relationship between risk scores and survival times, and distribution of risk scores in the TCGA test cohort (**A**) and in the TCGA−BRCA cohort (**B**). Kaplan−Meier curves of overall survival of BC patients in the TCGA test cohort (**C**), in the TCGA−BRCA cohort (**D**), and in the GSE20685 cohort (**E**) by the log−rank test. Time−dependent ROC curves and AUC analyses in the TCGA test cohort (**F**), in the TCGA−BRCA cohort (**G**), and in the GSE20685 cohort (**H**).

**Figure 3 molecules-28-03283-f003:**
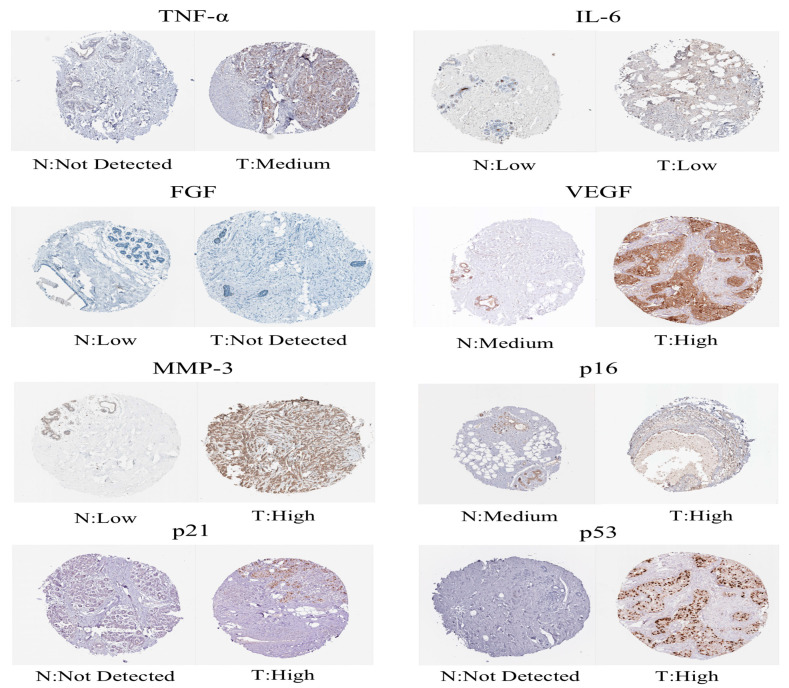
Immunohistochemical images obtained from the Human Protein Atlas.

**Figure 4 molecules-28-03283-f004:**
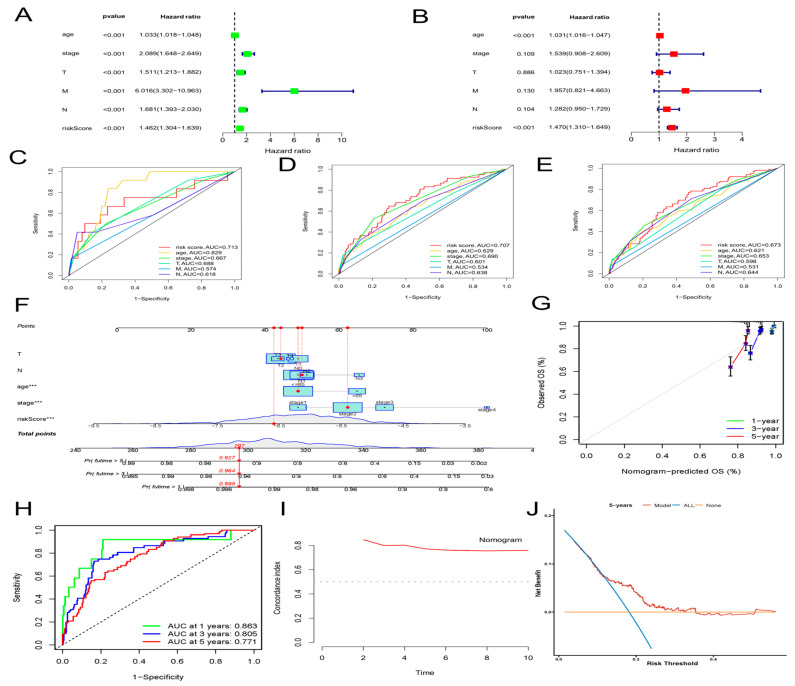
Construction and validation of a clinical prognostic nomogram for survival prediction. (**A**) Results of the univariate Cox regression analyses. (**B**) Results of the multivariate Cox regression analyses. (**C**–**E**) 1-, 3-, and 5-year ROC analyses of risk scores and clinical features. (**F**) The nomogram is based on the TCGA-BRCA cohort. (**G**) Calibration curves of the nomogram. (**H**) Time-dependent ROC curves and AUC analyses. (**I**) Concordance index (C-index) of the nomogram. (**J**) Decision curve analyses (DCA) comparing the predictive capacities.

**Figure 5 molecules-28-03283-f005:**
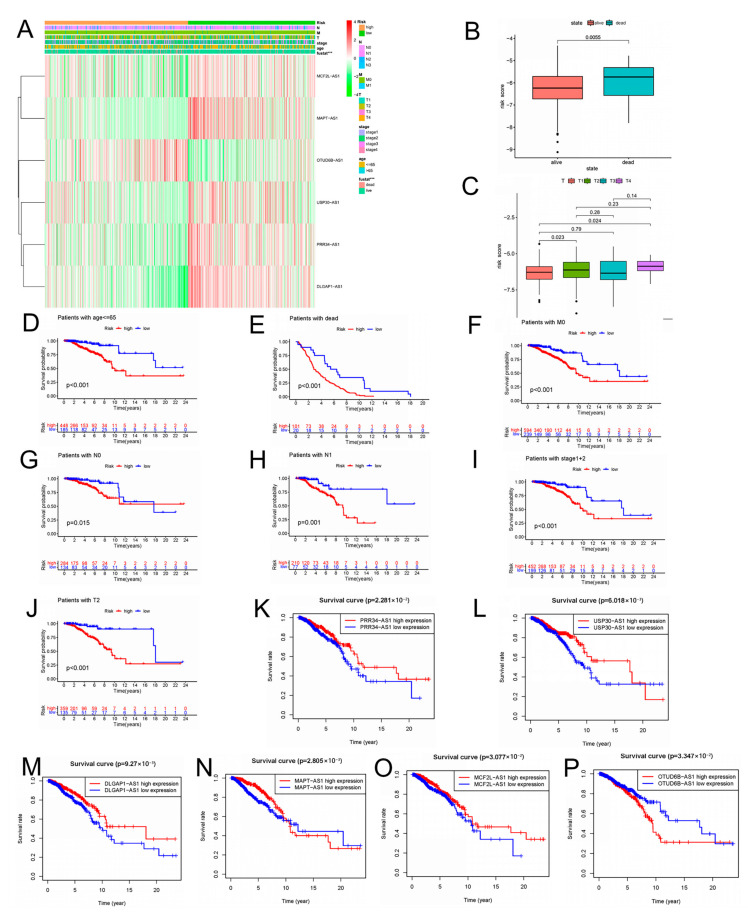
The relationship between risk scores and clinicopathological features. (**A**) The heatmap shows the distribution of six aging-related lncRNAs and clinicopathological features in high- and low-risk groups in TCGA-BRCA cohort. (**B**) Aging-related lncRNAs in the cohorts stratified by survival outcome. (**C**) Aging-related lncRNAs in the cohorts stratified by tumor size. (**D**–**J**) Correlation between different clinicopathological parameters and prognosis. (**K**–**P**) The expression of the six aging-related lncRNAs is significantly different in high- and low-risk groups using the log-rank test, *** *p* < 0.001.

**Figure 6 molecules-28-03283-f006:**
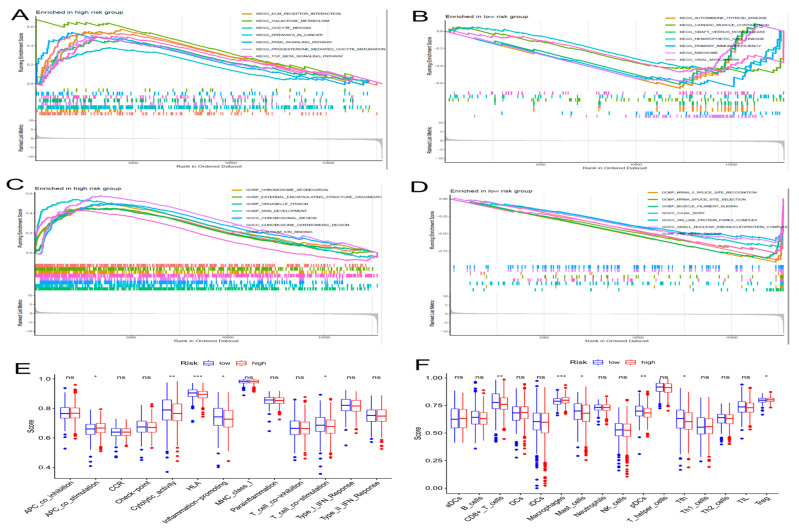
The gene set enrichment analysis (GSEA) and immune function between the high- and low-risk groups. The GSEA method was performed to identify and visualize the differential Kyoto Encyclopedia of Genes and Genomes (KEGG) (**A**,**B**) and Gene Ontology (GO) (**C**,**D**) enrichment analyses in the high- and low-risk groups. (**E**) Results of immune functions. (**F**) Results of infiltrating fractions of immune cells. * *p* < 0.05, ** *p* < 0.01, *** *p* < 0.001, ns (no significance).

**Figure 7 molecules-28-03283-f007:**
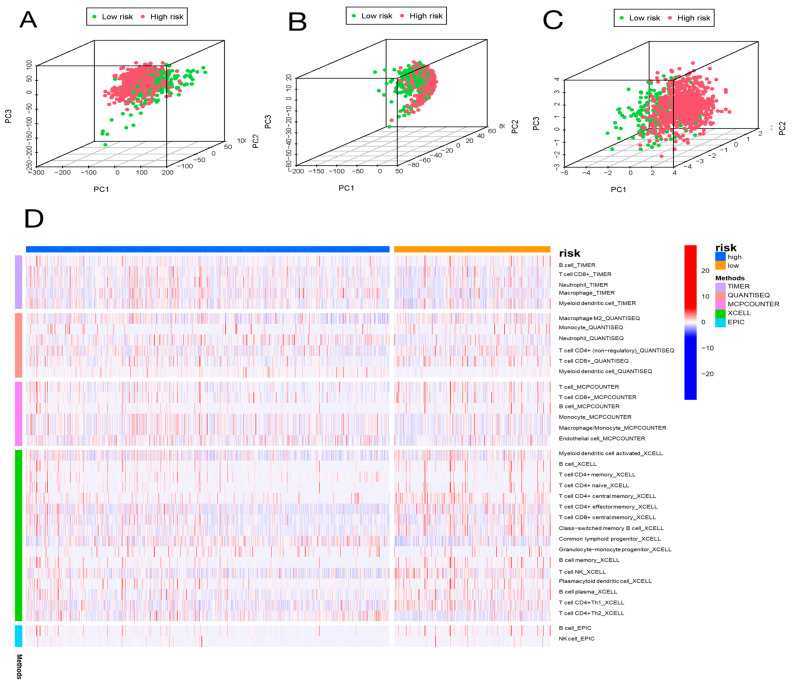
Principal component analysis (PCA) and correlation of risk scores with immune cell infiltration in different immune databases. PCA of low-risk and high-risk groups based on whole-genome (**A**), aging-related lncRNAs (**B**), and the risk signature including six aging-related lncRNAs (**C**). Patients with high risk scores are represented by red dots, patients with low risk scores are indicated by green dots. (**D**) The heatmap shows the correlation of risk scores with immune cell infiltration in different immune databases.

**Figure 8 molecules-28-03283-f008:**
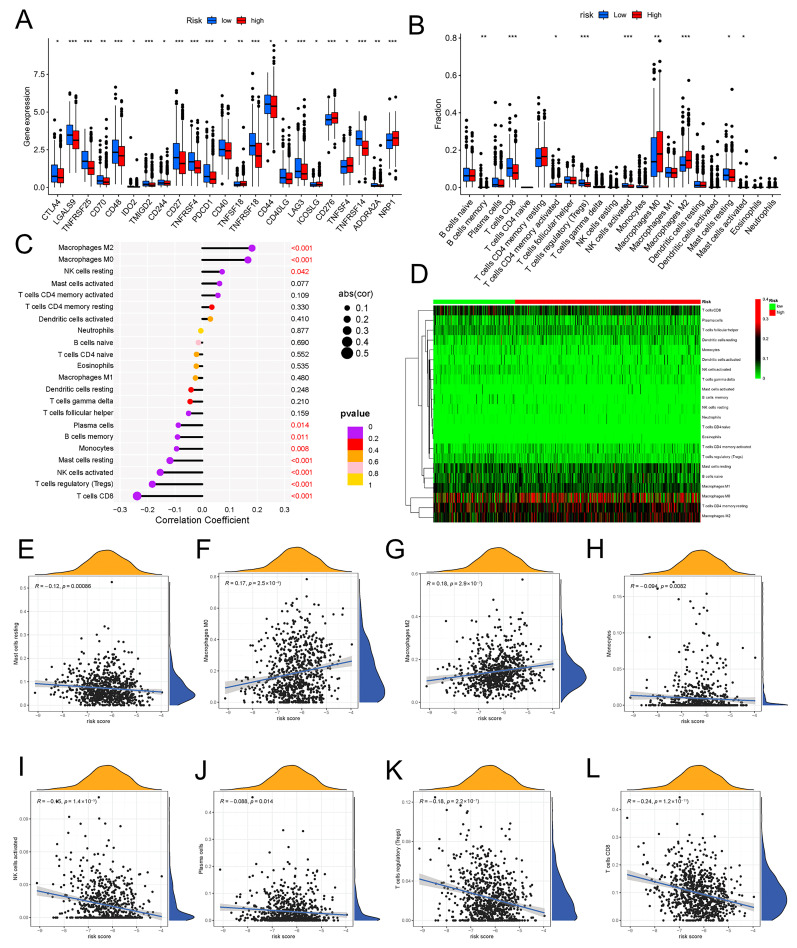
Expression of immune checkpoint molecules and the relationship between risk scores and immune cell infiltration in two different risk groups. (**A**) The distribution of immune checkpoints was significantly different between the high- and low-risk groups as determined by Wilcoxon test. (**B**) Differential analyses of the proportion of immune cell infiltration. Lollipop chart (**C**) and heatmap (**D**) showing correlation between risk scores and immune cell infiltration by Spearman rank correlation test. (**E**–**L**) Estimation of risk score coefficients for 8 types of immune cells using Spearman rank correlation test. * *p* < 0.05, ** *p* < 0.01, *** *p* < 0.001.

**Figure 9 molecules-28-03283-f009:**
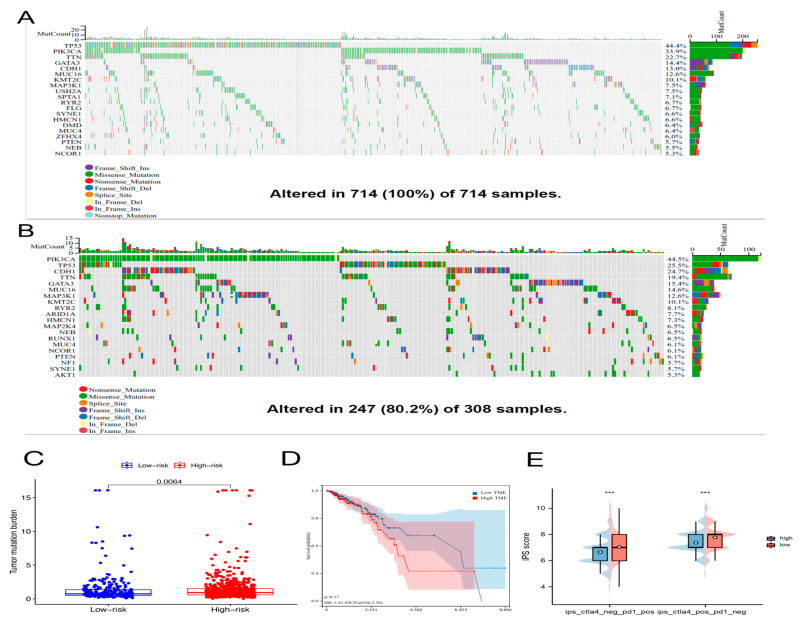
The mutation profile, tumor mutational burden (TMB), and immunophenoscore (IPS) in high-risk and low-risk groups. Mutation profile of BC patients in high-risk group (**A**) and in low-risk group (**B**). (**C**) The difference of TMB between two different risk groups as determined by the Wilcoxon test. (**D**) The relationship between TMB and overall survival of BC patients using the log-rank test. (**E**) Results of IPS in high-risk and low-risk groups using the Wilcoxon test, *** *p* < 0.001.

**Figure 10 molecules-28-03283-f010:**
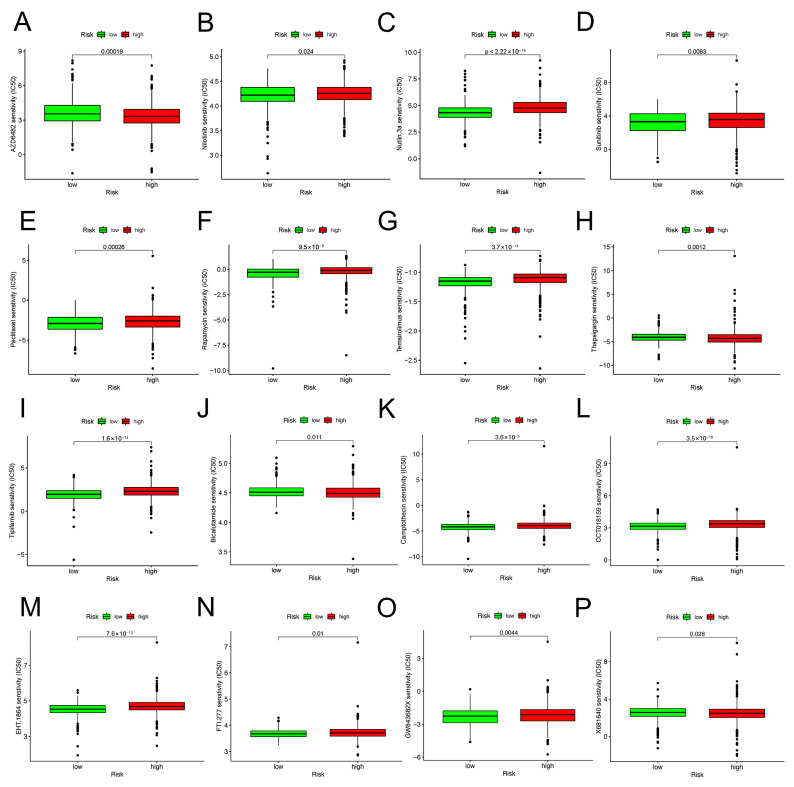
Drug sensitivity predictions as determined by the pRRophetic algorithm in R using the Wilcoxon test. (**A**–**P**) Different drug sensitivities associated with 6-ARlncRNAs signature.

## Data Availability

All data analyzed in the present study are publicly available in the Cancer Genome Atlas (TCGA; https://portal.gdc.cancer.gov/repository (accessed on 13 January 2023)), Gene Expression Omnibus (GEO; https://www.ncbi.nlm.nih.gov/geo/ (accessed on 13 January 2023)), and Human Protein Atlas (HPA; https://www.proteinatlas.org/ (accessed on 13 January 2023)).

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
