# Peer review of "A Novel Aging-Related Prognostic lncRNA Signature Correlated with Immune Cell Infiltration and Response to Immunotherapy in Breast Cancer"

_molecules, 2023, doi:10.3390/molecules28083283_

Round 1

Reviewer 1 Report

Dear authors:

This manuscript is well written and results are well supported by data and conclusion. Only a few questions need to be addressed.

1. Manuscript requires to be edited by native English speaker for English language and grammar.

2. Figure 3: Critical information regarding IHC images should be added, such as antibody info and staining intensity, etc. 

Reviewer 2 Report

The manuscript of Zhixin Liu is devoted to development of new prognostic method to monitor breast cancer progression. The manuscript is written by clear language and done at a high level. Nevertheless, I have some minor comments and suggestions.

1.      Abstract: Please give the full description what is lncRNA and AUC

2.      Please, designate in the corresponding places in the Methods Section and in each Figure legends, what tests for statistical analysis did the authors use.

3.      I suggest to move the first paragraph of the Discussion Section to the Introduction.

4.      Line 180: what is R? Please, give the full name of the used program.

Reviewer 3 Report

Liu, et al reports an interesting result that six novel aging related prognostic lncRNA signature correlated with immune cell infiltration and related to overall survival and response to immunotherapy in breast cancer patients. It is an important result that can provide treatment guide for breast cancer patients. However, there are some parts need to be clarified.

1.      We know the overall survival and treatment response is related to different breast cancer type. Does your lncRNA signature related to ER, PR and HER2 expression?

2.      In Materials and methods section, line 108, please explain the reason to use risk scoring formula. Same in Result section, line 215, where the number, eg (-0.6581)…..(-0.9132….., from?

3.      On line 225~226, it would be better to change the sentence to “in the two different risk groups, BC patients with higher risk score showed lower overall survival (Figure 1H)”

4.      In section 3.7, does the TMB related to immune checkpoint inhibitors treatment response?

5.      In section 3.8, line 407, how to define IPS-PD-1/PD-L1 pos and IPS-CTLA-4 post? >1%, >5%, >10%? CPS or TPS? Please make it clear

6.      Please do not use abbreviation when it appear as the first time in your manuscript. For example, line 84, GTF file, etc.

7.      Does gene signature which associated with survival of BC patients, has correlation with your lncRNA signature?
